# Promoting positive parenting and mental wellbeing in Hong Kong Chinese parents: A pilot cluster randomised controlled trial

**Yuying Sun[1], Man Ping Wang[2]\*, Christian S. Chan[3], Daphne L. O. Lo[4], Alice N. T. Wan[1]¤, Tai Hing Lam[1], Sai Yin Ho[1]**

**1** School of Public Health, The University of Hong Kong, Hong Kong SAR, China, **2** School of Nursing, The University of Hong Kong, Hong Kong SAR, China, **3** Department of Psychology, The University of Hong Kong, Hong Kong SAR, China, **4** The Hong Kong Family Welfare Society, Hong Kong SAR, China

¤ Current address: Aberdeen Kai-fong Welfare Association Social Service Centre, Hong Kong SAR, China
\* mpwang@hku.hk

## Abstract

### Objective

Effective and brief positive parenting interventions could be adopted widely, but evidence is limited. We aimed to evaluate the effectiveness of a positive parenting programme in Hong Kong Chinese parents.

### Methods

We conducted a pilot cluster randomised controlled trial in 2017 in 144 Hong Kong Chinese parents (84.7% women, mean age 42.5 [SD 5.87] years) of school-age children (mean age 10.9 [2.8] years) in 4 family service centres (clusters). The intervention included two 2-hour interactive talks (4 hours in total). The contents covered skills of giving praise, showing appreciation and playing enjoyable family games. The control group was offered the intervention after all the data were collected. Praise, appreciation and enjoyment related behaviours were measured as primary outcomes at baseline, 1 month and 3 months. The secondary outcomes were subjective happiness, wellbeing, personal health and happiness, family health, family happiness and harmony, and family relationship. After the completion of all assessments, five focus group discussions with the parents and four individual in-depth interviews with community service providers were conducted to explore their experiences.

### Results

Compared with the control group (n = 69), the intervention group (n = 75) showed greater positive changes in appreciation and enjoyment at 3 months with small effect sizes ($d$ = 0.42 and 0.32, respectively), and greater improvements in the secondary outcomes at 3 months with small effect sizes ($d$: 0.29–0.48). In the focus groups, the parents reported more praise to their children, better temper control, more focus on their children's strengths and better

**Data Availability Statement:** Data cannot be shared publicly because the informed consent explicitly stated that the individual data collected will only be available to the research team. The data

contains potentially identifying information including direct identifiers (contact information) and indirect identifiers (working status, income, etc.), which was restricted by the participant consent approved by Institutional Review Board of the University of Hong Kong/Hospital Authority Hong Kong West. Therefore, we are ethically unable to upload the raw dataset onto any publicly available websites. Data requests can be sent to the Mental Health Project (Community-based Mental Wellness Project for Adolescents and Adults), G/F, Patrick Manson Building, 7 Sassoon Road, Pok Fu Lam, Hong Kong. Email: sph.mhp@gmail.com.

**Funding:** This study was funded by Health and Medical Research Fund Health Care and Promotion Scheme (CPP-HKU) (awarded to THL, SYH and MPW). The funder had no roles in study design, data collection and analysis, decision to publish, or preparation of the manuscript."

**Competing interests:** The authors have declared that no competing interests exist.

family relationships. According to the service providers, most of the parents enjoyed the activities.

## Conclusions

The brief intervention in community settings with the engagement of community service providers has shown preliminary effectiveness in promoting positive parenting and mental wellbeing of Hong Kong Chinese parents.

## Trial registration

The authors confirm that all ongoing and related trials for this intervention are registered. The study reported in this manuscript is registered as clinical trial at clinicaltrials.gov: NCT03282071. https://clinicaltrials.gov/ct2/show/NCT03282071.

## Introduction

"Guan" (training, control or governance) possesses a very positive connotation in Chinese culture, meaning governing and training but also showing care and love [1]. Chinese parents value governance and obedience, which is rooted in culture under the influence of Confucianism [2]. Western parents tend to use praise in abundance in order to boost their children's confidence, whereas Chinese parents are apt to apply praise sparingly to prevent their children becoming complacent [3, 4]. A lower degree of acceptance and warmth were reported in Chinese parents, and also more hostile and neglecting compared with parents from other cultures [5]. Although training and controlling are considered positive in China [1], the high prevalence of child maltreatment is worrying. A 2006 report on Chinese parents with children aged 18 years or below found more than half of Hong Kong parents had used corporal punishment and nearly 5% of parents had maltreated their children physically, resulting in injuries [6]. Another report in 2019 on Hong Kong children attending Grade 1 to 3 (aged 6–10 years) showed that the past year prevalence of minor physical abuse, severe abuse, psychological abuse, and neglect were 64%, 23%, 84%, and 23%, respectively [7]. A meta-analysis of 22 studies in Chinese families showed that physical abuse was associated with adverse mental health outcomes [8]. In contrast, appreciation, warmth, affection and positive engagement predict children's social competence and school achievement [9, 10].

To reduce the risk of child maltreatment in the community, parental competence and parenting practices are to be enhanced. Parents may not be competent to praise children appropriately. Person praise (e.g., "You're smart!") and inflated praise (e.g., "That's incredibly beautiful!") may weaken children's motivation and feelings of self-worth [4]. Process praise, on the other hand, focuses on the efforts paid or the process (e.g., "Great job! You must have worked very hard."), which can enhance the recipients' intrinsic motivation when experiencing subsequent failure, and strengthen their resilience in facing difficulties [11]. Parents may benefit from training on positive parenting since they experience various child-rearing challenges. Low household income and education [12], low family functioning [13], children's behavioural problems [12] and high expectation towards children [14] were all the sources of parental stress. Parents who held stronger traditional Chinese values were found to have more feelings of shame towards child behaviour problems, and lower intentions to seek help [15]. The limited resources of services and potential stigma may be barriers to access to

consultation. Population-based approach is recommended to extend the impact of evidence-based interventions at the population level [16, 17]. A successful example is the multi-level Triple P model, using preventive parenting and family support strategy to enhance family protective factors and reduce risk factors associated with maltreatment [18, 19].

Brief and inexpensive interventions, if effective, can be adopted widely and sustainably [20]. Given the typical busy urban lifestyle in Hong Kong, brief interventions may encourage attendance and result in high adherence, especially for the general population who are self-perceived healthy. Only a few studies using less intensive positive parenting interventions were conducted among Chinese parents [20–23]. All of these studies adopted at least four group sessions of intervention [20–23]. Triple P model was implemented in Hong Kong by the Department of Health as part of its child health service. The programme was delivered by nurses who were supervised by a clinical psychologist, which involves substantial cost to the government and time to the parents [24]. Therefore, shorter community-based public health interventions are worth exploring, but relevant evidence is limited.

In January 2016, the Centre for Health Protection of the Department of Health of the Hong Kong Government launched a 3-year territory-wide "Joyful@HK" campaign to promote mental wellbeing under three themes: Sharing, Mind, and Enjoyment (SME) [25]. SME was used as the slogan of the Joyful@HK campaign. 'Sharing' connects family and friends, and supports those in need. 'Mind' entails keeping an open mind and being positive and optimistic. 'Enjoyment' is about engaging in enjoyable activities to maximise one's potential and achieve satisfaction [25]. Target populations of this campaign included adolescents, adults and elderly people. The projects in adults aimed to improve the public awareness of Mixed Anxiety and Depressive Disorder (MADD), which was the most common mental disorder in Hong Kong adults (6.9%) [26]. The present trial, Joyful Parenting Pilot Project, was one of the projects in adults under the campaign. With the participation of the community service providers, the objective of this pilot trial was to evaluate the effectiveness of simple interventions with only two sessions in promoting positive parenting behaviours and parents' wellbeing. We aimed to use enjoyable and simple family games to engage the parents and promote their practice of giving praise and showing appreciation. We hypothesised that participants in the intervention group would report more positive changes in giving praise and showing appreciation, as well as a higher level of happiness and wellbeing. Qualitative data were also collected through focus groups in parents and in-depth interviews in community service providers to understand the experiences of parents.

## Materials and methods

This pilot cluster randomised controlled trial used both quantitative and qualitative evaluation methods. The trial (NCT03282071) was conducted in 2017–18 by the University of Hong Kong and the Hong Kong Family Welfare Society (HKFWS). The organization supports holistic approach that encompasses the physical, mental and spiritual needs of different family members. For each integrated family service centre of the HKFWS, some clients use the services frequently and many of them may know each other well. Therefore, cluster design (assigning the participants recruited by the same centre to the same group) was adopted to avoid the potential contamination between the intervention and control groups.

### Participants

Four integrated family service centres of the HKFWS participated in the trial. Parents aged 18 to 59 years were recruited by the four service centres through leaflets and a public poster in the centres. Parents who were able to speak Chinese and complete questionnaires were eligible.

Those who could not read Chinese or had severe mental illness were excluded. Grandparents or other types of caregivers were not included. Family members (including children) of the participating parent were also invited to join a family gathering activity after completion of the interventions. Ethical approval was obtained from the Institutional Review Board of the University of Hong Kong / Hospital Authority Hong Kong West Cluster (reference number: UW17-240, dated 5 July 2017). Written informed consent was obtained in adult participants and parents provided written consent for children under the age of 18. The study was registered (NCT03282071, S1 Protocol). We reported the results following CONSORT guideline (S1 Checklist).

## Procedures

Four random numbers were generated by a computer. One person not involved in the randomisation process prepared sequentially numbered, opaque, and sealed envelopes, each containing a group allocation card. Two centres were randomised into the intervention group and two into the control group. The randomisation process was concealed from the researchers and cluster representatives. However, the recruitment staff and the enrolled parents were not blind to the allocation status as the intervention was obvious. We measured all the outcomes at baseline (before the first session of intervention), conducted the second assessment at 1 month (before the second session of intervention) and the third assessment at 3 months. A family gathering activity was provided to the participants after the final outcome assessment, which was used to encourage the participation and thank the participants for their completion of questionnaires. The waitlist control group received related services after data collection had been completed.

## Joyful parenting intervention

The design of the study was based on the concept of SME, Seligman's PERMA (Positive Emotion, Engagement, Relationships, Meaning and Accomplishment) model [27], and the findings from our previous project about increasing appreciation and reducing criticism in parents [28]. The interventions were designed by academic researchers together with community service providers, including two 2-hour interactive talks (4 hours in total) (S1 File). The contents included skills of giving praise, showing appreciation, and playing enjoyable and interactive family games. The instructors in each centre were two experienced social workers in the field of family counselling, with a senior social worker monitoring and providing guidance in the whole period.

In the first interactive talk, the instructors briefly introduced the symptoms of MADD and the ways to seek help for emotional disturbances. Then the instructors brought the importance of positive mind and discipline to build a positive environment and promote parent-child relationship. Also, SME and the benefit of praise and appreciation were introduced. Three simple and interactive family games were led by the social workers, enabling the parents to observe and experience how to give praise or show appreciation. The parents tried to praise each other through playing the games cooperatively. The principles of praise were emphasised: provide detailed and specific praise by using more adjective words (specific praise), praise for improvements rather than focusing on the achievements (give outcome praise in the right way: e.g. The parents had a high expectation on children's academic performance, but children's improvement from grade C to B was still worth praising), praise for their efforts made in the process even if they fail (process praise), have a consistent and reasonable standard, and be sincere. Parents were encouraged to write down eight strengths of their children and share with the other parents in the group. Simple worksheets were assigned to the parents to record their

practice of three types of praise (specific praise, process praise and outcome praise) and family games they arranged in the following four weeks. Times of praise (each type of praise done in one day was counted as 1 time) and family games in each week were documented.

After one month, we conducted the second interactive talk to reinforce the effect. The one-month gap between the two talks was to let the parents have time to practice the learned skills. They discussed about their experience in the past month and shared with the other participants. If the parents could not think of any words to praise, they could show their care and encouragement by simple actions such as a pat on the back. Lego games and role-playing games were organised in group format. Same as the ones in the first talk, these games were designed to remind the parents of the person and the behaviours that were worth praising. The parents were provided with the materials of the Lego and were encouraged to play together with their children at home and show appreciation during the games.

## Outcome measurements

**Primary outcomes: Praise, appreciation and enjoyment related behaviours.** We developed the outcome and impact-oriented questionnaires to assess the changes in the participants. Praise, appreciation and enjoyment were consistent with our delivered content of the intervention. Praise included 3 items (congeneric reliability was 0.88, range 0–21), "In the past seven days, how many days did you praise for children's efforts in words or actions / praise for children's strengths in words or actions / encourage children in words or actions". Each item of praise also included one sub-question about the frequency of praise each day (once, twice, three times, four times or more). The total times of praise in the past 7 days were calculated by multiplying the number of days and the times per day (range 0–84). Appreciation and Enjoyment in the past 7 days both included 2 items (range 0–14): "In the past seven days, how many days did you observe children's strengths carefully / understand children in a positive way", and "In the past seven days, how many days did you enjoy the time with children / hold outdoor activities with children". The congeneric reliability was 0.89 and 0.62, respectively. A higher score indicated higher level of positive parenting behaviour. Significant correlations were found between praise, appreciation, enjoyment and subjective happiness, wellbeing ($r = 0.19$–$0.43$, all $p < 0.05$), indicating that these measurements have acceptable validity in estimating better wellbeing and higher level of subjective happiness.

## Secondary outcomes

**Subjective happiness.** The 4-item Subjective Happiness Scale was used to assess individual participant's overall happiness [29]. The response of each item was a 7-point Likert scale. The score was the average of 4 items after reverse coding of the 4th item (range: 1 to 7). Higher scores indicated higher levels of happiness. The reliability and validity of the Chinese version have been established in the general population [30]. The Cronbach's alpha was 0.82, and the test-retest reliability was 0.70.

**Wellbeing.** The 7-item Short Warwick-Edinburgh Mental Well-being Scale with the 5-point Likert scale (1 = none of the time, 5 = all the time) was used. The score was calculated by summing all seven items with a range of 7 to 35 [31]. A higher score indicated higher level of wellbeing. The Chinese version indicated good validity and reliability in our previous paper [32]. The congeneric reliability was 0.85 and the test-retest reliability was 0.70.

**Personal health and happiness.** Personal health and happiness were measured by asking the respondents "How healthy / happy do you think you are". Respondents rated each item from 0 (not at all) to 10 (very healthy / happy). Higher scores indicate more healthy and happy [33].

**Family health, happiness and harmony.** Family health, happiness, and harmony were measured by asking the respondents "How healthy / happy / harmonious do you think your family is", which was reported in our previous papers [34, 35]. Respondents rated each item from 0 (not at all) to 10 (very healthy / happy / harmonious). Higher scores indicate higher level of family health, happiness, and harmony.

**Family relationship.** Family relationship included three items "the level of understanding / intimacy / communication with family members". Respondents rated each item from 0 (none) to 10 (full understanding / intimacy / communication with family members), resulting in a total score of 0–30. Higher scores indicate better family relationship. The congeneric reliability was 0.91.

## Intention to change

The intention to praise children's effort in words or actions, observe children's strength carefully, and enjoy the time with children were measured after the first talk. The questions were rated on a 5-point Likert scale, with "1" indicating "no intention at all" and "5" indicating "with strong intention".

## Subjective changes and process evaluation

The subjective changes in praise, appreciation and enjoyment were measured at 1 month and 3 months. The questions were rated on a 5-point Likert scale, with "1" indicating "much less" and "5" indicating "much more". Process evaluation was conducted after each talk. Some questions were rated on a 0–10 scale ("0" indicating "unsatisfied", "10" indicating "satisfied"), such as the satisfaction of the activity, whether they can learn SME from the activity, and whether they can gain mental health knowledge from the activity. The participants were also asked whether they would like to share the activity with others (yes / no).

## Evaluation of children

At 3 months, the children were invited to complete a short questionnaire by themselves or with the help of their parents. Their self-perceived praise, appreciation and enjoyment were asked: "how many days did your parents praise for your efforts in words or actions / praise for your strengths in words or actions / encourage you in words or actions / observe your strengths carefully / understand you in a positive way / enjoy the time with you / hold outdoor activities with you?". Questions about personal health and happiness, family health, happiness and harmony were also asked, which were the same as the parent's version.

## Focus groups and in-depth interviews

After completing all the quantitative outcome assessments, focus group discussion of the parents and individual in-depth interviews of the service providers were conducted. The focus group discussion was mainly to explore parents' satisfaction with the contents, subjective changes and suggestions for future programmes. The in-depth interviews aimed to explore community partners' perceptions on the usefulness, difficulties in implementation and suggestions. All discussions and interviews were conducted by one moderator (project coordinators who were familiar with the trial and had observed all the activities) and one note-taker. All the parents in the intervention group were invited to join the focus groups. Parents who were interested and available could join the discussions. Five focus group discussions of parents were conducted, with an average of 9 parents in each group (7–12 parents per group, 34

women and 11 men). Four social workers (1 man and 3 women) who have participated into the project were invited to complete the individual in-depth interviews.

## Fidelity

To evaluate whether the interventions had been delivered following the protocol, the researchers (project staff, government research officers, and senior officers from HKFWS) completed the fidelity checklist for each session of the activities, checking the extent that the actual activity aligned to the proposed rundown and the extent that the instructors conveyed the core messages.

## Data analysis

Quantitative data were analysed using STATA 13.0. Baseline characteristics were compared using Chi-square tests. Multilevel mixed-effects linear regression model (command XTMIXED) was used to calculate between-group mean differences (intervention vs. control) in the outcome changes, adjusting for clustering effect, significantly different demographics and baseline outcome variables. The model was fitted via restricted maximum likelihood method. The principle of intention-to-treat (ITT) analysis was adopted by including all the randomised subjects. The missing observations from lost to follow-up or not completing follow-up questionnaires were dealt with chained equations imputation [36]. Five imputed datasets were generated and pooled using Rubin's rules [36]. An effect size (Cohen's d) of 0.2 was considered as a small effect, 0.5 a medium effect, and 0.8 a large effect [37]. All significance tests were two-sided with a 5% level of significance. A supplementary analysis was conducted by only adjusting for clustering effect and baseline outcome variables, which was unspecified in the protocol. Qualitative data were analysed using thematic analysis [38]. All the interviews were transcribed verbatim, and the transcripts were read throughout. The relevant or interesting keywords were highlighted as initial codes. The codes were collated into potential themes and the relevant data were gathered together. The entire data set was reviewed, and the themes were checked again to figure out a thematic map. The themes were checked and refined until clear definitions and names were created. Some compelling contents were extracted and reported as examples [38].

## Results

Four family centres (144 parents, mean age 42.5 (SD 5.87) years) were recruited with two centres randomised into the intervention group (75 parents, mean age 41.8 (5.51) years) and two into the control group (69 parents, mean age 43.3 (6.19) years) in September 2017. Fig 1 shows the CONSORT flow chart. The retention rates of the intervention group and control group were 96% at 1 month and 87% at 3 months. Table 1 shows the majority of the participants (77.3% in the intervention group and 92.8% in the control group, $p = 0.010$) were women. The intervention group had higher education level ($p = 0.001$), higher proportion in full-time employment ($p = 0.017$), and higher household income ($p = 0.03$) than the control group. Sex, education level, working status, household income and baseline of the corresponding outcome variable were included as covariates in subsequent outcome analyses.

### Primary outcomes

The parents in the intervention group showed greater positive changes in appreciation (between-group mean difference, BMD = 1.19, 95% CI: 0.27 to 2.12, Cohen's $d$ = 0.42, $p = 0.011$) and enjoyment (BMD = 0.98, 0.01 to 1.94, $d$ = 0.32, $p = 0.047$) than the control group at 3 months (Table 2). Praise did not show significant changes at 3 months (Table 2). S1

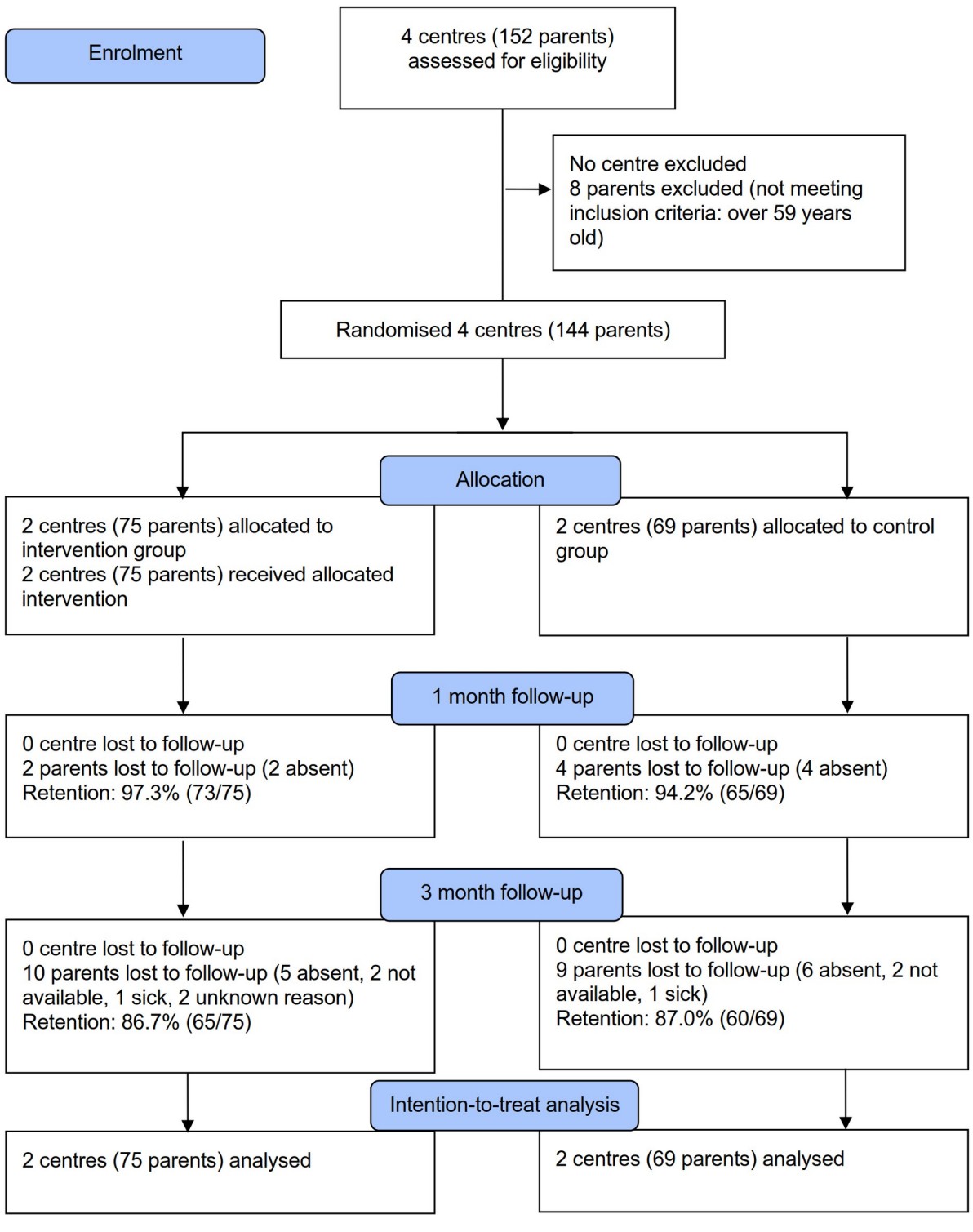

**Fig 1. CONSORT flow chart.**

Table shows the outcomes without adjusting for baseline characteristics, which were similar to the adjusted outcomes.

**Table 1. Demographic characteristics.**

| Demographics | Categories | Total (n = 144) n (%) | Intervention (n = 75) n (%) | Control (n = 69) n (%) |
|---|---|---|---|---|
| Sex | Man | 22 (15.3) | 17 (22.7) | 5 (7.2) |
| | Woman | 122 (84.7) | 58 (77.3) | 64 (92.8) |
| Birthplace | Hong Kong | 55 (38.2) | 28 (37.3) | 27 (39.1) |
| | Guangdong Province | 56 (38.9) | 29 (38.7) | 27 (39.1) |
| | Other places | 33 (22.9) | 18 (24.0) | 15 (21.7) |
| Marital status | Married | 122 (84.7) | 66 (88.0) | 56 (82.4) |
| | Unmarried | 22 (15.3) | 9 (12.0) | 13 (18.8) |
| Education level | Primary or below | 7 (16.7) | 1 (1.3) | 6 (8.7) |
| | Secondary or diploma | 133 (78.5) | 55 (73.3) | 58 (84.1) |
| | Degree or higher | 24 (16.7) | 19 (25.3) | 5 (7.2) |
| Working status | Full-time work | 40 (27.8) | 27 (36.0) | 13 (18.8) |
| | Part-time work | 21 (14.6) | 11 (14.7) | 10 (14.5) |
| | Housekeeper | 75 (52.1) | 34 (45.3) | 41 (59.4) |
| | Others | 8 (5.6) | 3 (4.0) | 5 (7.2) |
| Household monthly income, HK$ | < 20,000 | 74 (51.7) | 31 (41.3) | 43 (63.2) |
| | 20,000 to 40,000 | 41 (28.7) | 27 (36.0) | 14 (20.6) |
| | > 40,000 | 28 (19.6) | 17 (22.7) | 11 (16.2) |
| Number of children | 1 | 43 (29.9) | 25 (33.3) | 18 (26.1) |
| | 2 | 79 (54.9) | 41 (54.7) | 38 (55.1) |
| | 3 or more | 22 (15.3) | 9 (12.0) | 13 (18.8) |

According to the collected worksheets of home practice from 52 parents, praise was given 7.13 (SD 5.21), 6.69 (5.79), 6.63 (5.83) and 4.48 (5.84) times from the first to the fourth week, respectively. There was a statistically significant difference between groups ($F$ (3, 204) = 4.101, $p = 0.007$). The post hoc test revealed that the home practice of praise was lower in the fourth week compared with the first ($p = 0.018$) and second week ($p = 0.048$). Parents arranged 2.87 (2.20), 2.73 (2.18), 2.46 (2.00) and 1.56 (1.99) times of family games from the first to the fourth weeks. The fourth week practice of family games reduced significantly compared with the first three weeks ($p = 0.002$, $p = 0.005$ and $p = 0.029$, respectively).

## Secondary outcomes

The parents in the intervention group showed greater increases in subjective happiness at 1 month (BMD = 0.36, 0.10 to 0.62, $d = 0.33$, $p = 0.006$) and 3 months (BMD = 0.39, 0.14 to 0.64, $d = 0.39$, $p = 0.002$) than the control group (Table 2). We also found greater and significant improvements in the other outcomes at 3 months, including wellbeing (BMD = 1.20, 0.17 to 2.24, $d = 0.29$, $p = 0.023$), personal health (BMD = 0.81, 0.26 to 1.35, $d = 0.41$, $p = 0.004$), personal happiness (BMD = 0.72, 0.21 to 1.23, $d = 0.37$, $p = 0.006$), family health (BMD = 0.65, 0.12 to 1.17, $d = 0.36$, $p = 0.016$), family happiness (BMD = 0.74, 0.22 to 1.25, $d = 0.41$, $p = 0.005$), family harmony (BMD = 0.72, 0.22 to 1.22, $d = 0.37$, $p = 0.005$) and family relationship (BMD = 2.36, 0.96 to 3.76, $d = 0.48$, $p = 0.001$) (Table 2).

## Intention to change

After the first talk, all the parents reported intention or strong intention to praise children's effort in words or actions, observe children's strengths carefully, and enjoy the time with children. Fifty-two percent to 61% parents reported strong intention to have more positive parenting behaviours.

**Table 2. The comparison between intervention group (n = 75) and control group (n = 69).**

| Items | Phase | Mean (SD) | | BMD (95%CI) [a] | Cohen's d (95% CI) | P-value [b] |
|---|---|---|---|---|---|---|
| | | Intervention (n = 75) | Control (n = 69) | Intervention vs Control | Intervention vs Control | |
| Praise (days) | Baseline | 11.49 (5.15) | 11.34 (4.79) | | | |
| | 1 month | 12.23 (4.91) | 11.05 (4.81) | 0.99 (-0.35, 2.34) | 0.20 (-0.07, 0.48) | 0.148 |
| | 3 months | 13.09 (4.12) | 12.34 (4.22) | 0.10 (-1.37, 1.57) | 0.02 (-0.33, 0.38) | 0.896 |
| Praise (times) | Baseline | 28.35 (22.92) | 25.56 (18.54) | | | |
| | 1 month | 28.87 (22.34) | 22.35 (16.88) | 5.24 (-0.01, 10.49) | 0.26 (-0.001, 0.53) | 0.050 |
| | 3 months | 28.61 (18.33) | 25.40 (16.72) | 1.46 (-2.88, 5.80) | 0.08 (-0.16, 0.33) | 0.510 |
| Appreciation | Baseline | 7.03 (3.72) | 7.86 (3.56) | | | |
| | 1 month | 8.37 (3.66) | 8.44 (3.36) | 0.06 (-1.04, 1.16) | 0.02 (-0.30, 0.33) | 0.918 |
| | 3 months | 8.86 (2.81) | 7.83 (2.91) | 1.19 (0.27, 2.12) | 0.42 (0.09, 0.74) | 0.011 |
| Enjoyment | Baseline | 7.78 (3.78) | 7.42 (3.29) | | | |
| | 1 month | 7.82 (3.22) | 7.58 (3.09) | 0.41 (-0.61, 1.43) | 0.13 (-0.19, 0.45) | 0.435 |
| | 3 months | 8.92 (3.31) | 7.75 (2.77) | 0.98 (0.01, 1.94) | 0.32 (0.003, 0.63) | 0.047 |
| Subjective happiness | Baseline | 4.47 (1.19) | 4.52 (1.10) | | | |
| | 1 month | 5.03 (1.10) | 4.54 (1.08) | 0.36 (0.10, 0.62) | 0.33 (0.09, 0.57) | 0.006 |
| | 3 months | 5.12 (0.93) | 4.64 (1.05) | 0.39 (0.14, 0.64) | 0.39 (0.14, 0.65) | 0.002 |
| Well-being | Baseline | 24.82 (3.80) | 23.82 (4.62) | | | |
| | 1 month | 25.81 (3.94) | 24.67 (3.74) | 0.31 (-0.69, 1.31) | 0.08 (-0.18, 0.34) | 0.538 |
| | 3 months | 27.32 (3.83) | 24.81 (4.55) | 1.20 (0.17, 2.24) | 0.29 (0.04, 0.53) | 0.023 |
| Personal health | Baseline | 6.15 (2.05) | 6.10 (2.27) | | | |
| | 1 month | 6.99 (2.04) | 6.18 (2.23) | 0.40 (-0.09, 0.89) | 0.19 (-0.04, 0.42) | 0.109 |
| | 3 months | 7.29 (1.84) | 6.32 (2.15) | 0.81 (0.26, 1.35) | 0.41 (0.13, 0.68) | 0.004 |
| Personal happiness | Baseline | 6.15 (2.31) | 6.15 (2.19) | | | |
| | 1 month | 6.97 (1.99) | 6.45 (2.11) | 0.25 (-0.21, 0.72) | 0.12 (-0.10, 0.35) | 0.281 |
| | 3 months | 7.25 (1.79) | 6.37 (2.07) | 0.72 (0.21, 1.23) | 0.37 (0.11, 0.64) | 0.006 |
| Family health | Baseline | 6.25 (2.32) | 6.25 (2.22) | | | |
| | 1 month | 7.23 (1.87) | 6.48 (1.94) | 0.39 (-0.09, 0.86) | 0.20 (-0.05, 0.45) | 0.110 |
| | 3 months | 7.33 (1.66) | 6.50 (1.95) | 0.65 (0.12, 1.17) | 0.36 (0.07, 0.65) | 0.016 |
| Family happiness | Baseline | 6.40 (2.54) | 6.37 (2.13) | | | |
| | 1 month | 7.38 (1.99) | 6.55 (1.92) | 0.46 (-0.04, 0.95) | 0.24 (-0.02, 0.49) | 0.070 |
| | 3 months | 7.45 (1.67) | 6.46 (1.98) | 0.74 (0.22, 1.25) | 0.41 (0.12, 0.68) | 0.005 |
| Family harmony | Baseline | 6.23 (2.59) | 6.49 (2.09) | | | |
| | 1 month | 7.20 (2.09) | 6.80 (1.96) | 0.27 (-0.23, 0.77) | 0.13 (-0.11, 0.38) | 0.294 |
| | 3 months | 7.39 (1.89) | 6.63 (1.96) | 0.72 (0.22, 1.22) | 0.37 (0.11, 0.63) | 0.005 |
| Family relationship | Baseline | 20.71 (6.80) | 20.85 (5.05) | | | |
| | 1 month | 22.36 (5.03) | 20.73 (4.76) | 1.11 (-0.16, 2.39) | 0.23 (-0.03, 0.49) | 0.087 |
| | 3 months | 23.06 (4.69) | 20.34 (5.23) | 2.36 (0.96, 3.76) | 0.48 (0.19, 0.76) | 0.001 |

BMD, between-group mean difference; CI, confidence interval.

[a] The differences between two groups at 1 month or 3 months were adjusted for the baseline of the corresponding variables, sex, education level, working status and family income, and cluster effect.

[b] p values were calculated using multilevel mixed-effects linear regression model.

## Subjective changes and process evaluation

At 1 month and 3 months, more than 95% parents perceived a little or much more positive changes of parenting behaviours. Most of the parents in the intervention group reported improvement of health (78.9%) and happiness (90.1%). The scores of the process evaluation at

different time-points were all above 8 out of 10 (ranged from 8.2 to 8.8) except one item "can learn from MADD" (score 7.9). All the parents reported the willingness to share the programme with others.

## Fidelity

The adherence to the proposed rundown of the first and the second interactive talk was 87.8% (SD 15.0) and 96.3% (7.4), according to the evaluation of 11 observers and 9 observers, respectively. The adherence to the core messages of the first and the second talk was 85.6% (7.3) and 90.0% (9.3), respectively.

## Evaluation of the children

A total of 141 children completed the questionnaires before the family gathering activity. The average age was 10.9 (2.8) years (range 4 to 18 years; 5.6% aged 4–7 years, 61.5% aged 8–11 years, 23.1% aged 12–15 years, 9.8% aged 16–18 years). Around half the children were boys (51.8%). Most of the children were in primary school (62.4%) or secondary school (32.6%). No significant difference in the outcomes of the children was found between the intervention group (n = 70) and the control group (n = 71).

## Focus groups and in-depth interviews

The major themes included the overall impression, the impact of the interventions and subjective changes, difficulties met during practice, and suggestions for future improvement. Parents thought that the programme was helpful. Most of the parents enjoyed the simple family games, which made them understand the praise skills soon after engaging into the activities. Through playing games, the interaction with other parents or families brought fun, joy and relaxation. Some parents liked the cohesion of learning and playing.

## Helpful programme in general

*"I like the games. The atmosphere was quite nice. During the process, the instructors taught many positive words to praise. Everyone enjoyed and felt happy." (Male, 46 years old)*

**Positive changes.** Parents reported positive changes after the intervention, such as more praise to their children and paying more attention to observe their children's strength. Their family relationship improved, and children performed better after they were given praise. Some parents reported better control of temper when their children did not perform well.

*"My child become more proactive and more willing to help me with the housework." (Female, 42 years old)*

*"I know more about praise and appreciation (after the talks). Our relationship has improved." (Female, 45 years old)*

*"We used to focus on success and winning all the time. Now I try to observe my child's strengths more carefully." (Female, 47 years old)*

*"When I get really angry and want to criticize my child, I try to control my temper." (Female, 37 years old)*

**Difficulties met.**    It was hard for the parents to control temper sometimes. They met some difficulties when praising their children in daily life, particularly for the older children. It was common that the parents got angry when their children did not perform well in school work.

*"My child did not focus on study even before the examination. I talked to him, but he would not listen. Then I got angry." (Female, 40 years old)*

*"It is quite hard to praise children as you must observe carefully all the time. Perhaps I'm not good at it. I could only use a few common words." (Female, 42 years old)*

*"My older child is a student in middle school, and he complained that my praise was a little bit annoying." (Female, 41 years old)*

**Suggestions.**    Some parents suggested that more similar workshops and talks should be arranged in the future. The programme could be delivered to the grandparents who were also the common caregivers of children.

*"It would be nice to have more similar classes, or we would forget all the contents after a while." (Female, 41 years old)*

**Community service providers' view.**    Community service providers said most of the parents enjoyed the activities. The social workers themselves also learned from the project. However, they felt some difficulties in delivering the heavy contents in two talks. They suggested that the intervention sessions should last longer.

*"The parents were very cooperative in the games. Most of them were involved and very happy." (Male)*

*"We often focus on changing parent's mindset to be positive. This activity made me realize that sometimes they could act first, even though they need some time to convert the way of thinking afterwards." (Female)*

*"I think the time is too rush, the activity was only two hours but included playing, teaching, and filling questionnaires." (Female)*

## Discussion

We evaluated the effectiveness of two simple interactive talks in improving the positive parenting behaviours in Hong Kong Chinese parents. The implementation had followed the protocol with high fidelity. The compliance to intervention was high at the follow-ups. The quantitative results showed the effectiveness of our intervention in improving parents' appreciation and enjoyment, with small effect sizes at 3 months ($d$: 0.42 and 0.32, respectively), which was our primary goal. Moreover, the subjective happiness, wellbeing, personal health and happiness, family health, happiness and harmony, and family relationship ($d$: 0.29–0.48) all improved with small effect sizes at 3 months. Primary prevention mental health programmes usually have small effect on positive parenting behaviours [39]. Our findings are consistent with the universal approach for parenting practices ($d$: 0.39) [19].

We did not observe significant changes in praise, according to the quantitative data. However, more than 95% parents reported that they expressed more praise and appreciation to their children. The qualitative data have also shown more praise in some parents. The small

sample size of the pilot study might have led to the non-significant increase of parents' praise behaviour. Another possible explanation is that the practice of praise and family games has just decreased at the one-month evaluation point. According to the collected worksheets of home practice, the 4[th] week practice of praise and family games reduced significantly, just within the period of our one-month outcome evaluation. Process praise required detailed description and information, which might be difficult for parents to practise. Some parents also reported difficulties in temper control, especially when they were not satisfied with their children's performance in schools. The mindset of fixing problems and belief in the usefulness of criticism may be difficult to change in short-term. We did not measure criticism or abuse in the present study, but it would be useful to include the measurements in future study, to evaluate whether such programs could prevent or reduce children maltreatment.

Although parents did not increase the frequency of praise, it is possible that the intervention has changed the mindset of the parents as evidenced by the higher level of appreciation and enjoyment at 3 months. The improvement in appreciation and enjoyment might lead to positive changes in subjective happiness, wellbeing, personal health and happiness, family health, happiness and harmony, and family relationship. The quality of appreciation and communication may be more important than the quantity of praise. To get a sustainable effect, regular booster interventions and activities are needed to further improve the quality of praise and appreciation. Our simple interventions and family games were to involve parents into the activities quickly. We emphasised the importance of practice and used simple messages to make them act immediately. The core components are clear to the participants but the skills to deal with different situations in daily life are to be strengthened.

Our study had several limitations. First, because validated questionnaires were not available and many existing ones were too long, we developed the outcome and impact-oriented questionnaires to assess the changes in the participants. The measurements showed acceptable to good congeneric reliability. Future studies should test the validity of the praise measurements. Second, we did not measure the outcomes of children at baseline, thus the comparison between the intervention and control group might be affected by the imbalanced baseline. However, the interventions and outcome evaluation mainly targeted on parents. Future studies could include more measurements on the children. Third, because the target population was healthy and the 'usual care' was no care, a waitlist control was adopted. To control for any effect due to attention or interaction, alternative psychological placebos could be used in future studies. Fourth, as a pilot trial, only four family centres were involved, and there might be some other unmeasured confounders. Although pilot studies would not aim at statistical significance, we did record some significant changes. The intervention can thus be scaled up and larger trials with improvements are warranted. Given the difficulties in recruiting fathers, and mothers may have a stronger influence on the children rearing in Hong Kong, targeting mothers seems to be an effective strategy.

Population-based interventions and primary prevention are scarce but necessary in the healthcare system. Our pilot trial of population-based positive parenting in Hong Kong Chinese parents, with the engagement of community service providers, can be a useful reference by future larger trials or adopted as a routine community practice. The skills of praise, appreciation and enjoyment are part of positive parenting and a set of parenting skills such as temper control and positive discipline might also be included in future programmes. The programmes for building parental competence not only help parents increase their own self-efficacy in positive parenting, but also benefit the children and prevent the children maltreatment.

## Supporting information

**S1 Protocol.**
(PDF)

**S1 Checklist. CONSORT 2010 checklist of information to include when reporting a randomised trial**∗**.**
(DOC)

**S1 File. Programme rundown.**
(DOCX)

**S1 Table. The comparison between intervention group (n = 75) and control group (n = 69).**
(DOCX)

## Acknowledgments

We would like to express our sincere thanks to the participants for their attention and active involvement, and all the social workers from the Hong Kong Family Welfare Society (HKFWS) for conducting the interventions.

## Author Contributions

**Conceptualization:** Yuying Sun, Man Ping Wang, Christian S. Chan, Daphne L. O. Lo, Alice N. T. Wan, Tai Hing Lam, Sai Yin Ho.

**Data curation:** Yuying Sun, Man Ping Wang, Alice N. T. Wan, Sai Yin Ho.

**Formal analysis:** Yuying Sun, Man Ping Wang, Sai Yin Ho.

**Funding acquisition:** Man Ping Wang, Tai Hing Lam, Sai Yin Ho.

**Investigation:** Yuying Sun, Man Ping Wang, Daphne L. O. Lo, Alice N. T. Wan, Sai Yin Ho.

**Methodology:** Yuying Sun, Man Ping Wang, Christian S. Chan, Daphne L. O. Lo, Alice N. T. Wan, Tai Hing Lam, Sai Yin Ho.

**Project administration:** Yuying Sun, Daphne L. O. Lo, Alice N. T. Wan.

**Resources:** Daphne L. O. Lo, Tai Hing Lam.

**Supervision:** Man Ping Wang, Christian S. Chan, Alice N. T. Wan, Tai Hing Lam, Sai Yin Ho.

**Validation:** Yuying Sun, Man Ping Wang, Christian S. Chan.

**Writing – original draft:** Yuying Sun, Man Ping Wang, Sai Yin Ho.

**Writing – review & editing:** Yuying Sun, Man Ping Wang, Christian S. Chan, Daphne L. O. Lo, Alice N. T. Wan, Tai Hing Lam, Sai Yin Ho.

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
