## [Decision Letter · Decision Letter 0]

25 Jun 2021

PONE-D-20-37785

Promoting positive parenting and mental wellbeing in Hong Kong Chinese parents: a cluster randomised controlled trial

PLOS ONE

Dear Dr. Wang,

Thank you for submitting your manuscript to PLOS ONE. After careful consideration, we feel that it has merit but does not fully meet PLOS ONE’s publication criteria as it currently stands. Therefore, we invite you to submit a revised version of the manuscript that addresses the points raised during the review process.

The reviewers have raised a number of concerns regarding the study design, methods and analyses that need to be thoroughly addressed in your revisions. Please ensure that you attend carefully to the points they have identified.

We look forward to receiving your revised manuscript.

Kind regards,

Jamie Males

Staff Editor

PLOS ONE

Journal Requirements:

”The funders had no role in study design, data collection and analysis, decision to publish, or preparation of the manuscript”

Reviewers' comments:

Reviewer's Responses to Questions

**Comments to the Author**

1. Is the manuscript technically sound, and do the data support the conclusions?

Reviewer #1: Partly

Reviewer #2: Partly

Reviewer #3: Partly

2. Has the statistical analysis been performed appropriately and rigorously? 

Reviewer #1: I Don't Know

Reviewer #2: I Don't Know

Reviewer #3: Yes

3. Have the authors made all data underlying the findings in their manuscript fully available?

Reviewer #1: No

Reviewer #2: No

Reviewer #3: No

4. Is the manuscript presented in an intelligible fashion and written in standard English?

Reviewer #1: Yes

Reviewer #2: Yes

Reviewer #3: Yes

5. Review Comments to the Author

Reviewer #1: Important note: This review pertains only to ‘statistical aspects’ of the study and so ‘clinical aspects’ [like medical importance, relevance of the study, ‘clinical significance and implication(s)’ of the whole study, etc.] are to be evaluated [should be assessed] separately/independently. Further please note that any ‘statistical review’ is generally done under the assumption that (such) study specific methodological [as well as execution] issues are perfectly taken care of by the investigator(s). This review is not an exception to that and so does not cover clinical aspects {however, seldom comments are made only if those issues are intimately / scientifically related & intermingle with ‘statistical aspects’ of the study}. Agreed that ‘statistical methods’ are used as just tools here, however, they are vital part of methodology [and so should be given due importance].

COMMENTS: In my opinion, sample size used is small because as per lines 36-38 [Results: Compared with the control group (n = 69), the intervention group (n = 75) showed greater positive changes in appreciation and enjoyment at 3-month with small effect sizes (d = 0.42 and 0.32, respectively)] the effect size is ‘small’ and small effect size study needs a larger sample size {refer to table-2, page-158 of: “A power primer” by J. Cohen in Psychological Bulletin, 1992, vol.:112, pp 155-159]. Considering that it is a ‘pilot’ (line 25), of course, sample size is not a big issue. [However, I very strongly feel that a word ‘pilot’ study should also appear in title.]

Since this is a [though pilot] cluster randomised controlled trial, statistical comparison of baseline characteristics [last ‘p-value’ column in Table 1] is not desirable. Please note

To provide a description of baseline characteristics is entirely reasonable (since it is clearly important in assessing to whom the results of the trial can be applied), however, it does not require the division of baseline characteristics by treatment groups (however, if done – alright). Statistical comparison of baseline characteristics is not desirable at all [because even if P-value turns out to be significant (while comparing baseline characteristics despite random allocation), it is, by definition, a false positive] as you then are supposed to be testing ‘randomization’ then, which in any single trial may not balance all baseline characteristics because ‘randomization’ is a sort of ‘insurance’ and not a guarantee scheme.

References:

1. Stuart J. Pocock, et al., ‘Subgroup analysis, covariate adjustment and baseline comparisons in clinical trial reporting: current practice and problems’, Statistics in medicine, 2002; 21:2917–2930 [Particularly page 2927]

2. Harrington D, et al., ‘New guidelines for statistical reporting in the journal’, N Engl J Med 2019;381:285-6

[Important message (indirectly/ultimately indicated) from these articles: Never do any comparison with respect to ‘baseline’ characteristics {by applying statistical significance test(s)}, when allocation is done randomly].

’P-values’ {last column} reported in Table 2 [The comparison between intervention group (n=75) and control group (n=69)] are supposed to have yielded by “Multilevel mixed-effects linear regression model” [as nothing has been given in the footnote]. It could have been simple to do it by non-parametric equivalent to unpaired ‘t’ test namely Mann-Whitney ‘U’ test on ‘change scores’ as though the measures/tools used are appropriate, most of them yield data that are in [at the most] ‘ordinal’ level of measurement [and not in ratio level of measurement for sure {as the score two times higher does not indicate presence of that parameter/phenomenon as double (for example, a Visual Analogue Scales VAS score or say ‘depression’ score)}]. Then application of suitable non-parametric test(s) is/are indicated/advisable [even if distribution may be ‘Gaussian’ (i.e. normal)]. Agreed that there is/are no non-parametric test(s)/technique(s) available to be used as alternative in all situation(s) [suitable / most desired/applicable], but should be used whenever/wherever they are available

Further, in my opinion, account given in ‘Focus groups and in-depth interviews’ section (lines 307-358) could be [and should be, I guess] reduced. Referring to lines 395-405 [Our study had several limitations…….] studies ‘quality’ becomes highly questionable. Even as a pilot study, in my considered opinion, this study does not contribute any new information.

Reviewer #2: Overall

This paper investigated the efficacy of a brief positive parenting programme on or positive parenting (praise, appreciation, enjoyment) in a sample of 144 Hong Kong parents. The study had a mixed-method design, combining a randomized controlled trial (RCT) with group discussions. Quantitative measures were taken at baseline and again 1 month and 3 months post intervention. Group discussions with parents and individual discussions with providers were held post-intervention. Intervention effects were found for positive parenting and mental wellbeing.

The promotion of positive parenting is of great importance, especially in a Hong Kong context, given the prevalence of “tiger parenting” within Hong Kong (and Chinese) communities. I commend the authors for undertaking this study and for drawing attention to the effects of positive parenting. Well done for recruiting 144 parents, for conducting implementation fidelity checks, and for controlling for confounders in the analysis.

Introduction

The authors provide a rationale for the study, although the rationale for the actual intervention is not convincing to me. The introduction needs revision as there is some unclarity and a lack of nuance. For example:

52: In Chinese culture, it is commonly thought that praising children breeds complacency and hinders learning

54: “more than half of Hong Kong parents has used corporal punishment and 4.5% has maltreated their child physically”

- I am confused by this sentence as, to me, corporal punishment is physical maltreatment. In my view, this sentence needs to be rephrased, looking carefully what the cited authors meant

- 56: The paper (5) that the authors cite is not focused on Chinese/ Hong Kong parents and is perhaps not the best reflection of Hong Kong parenting and child outcomes. Cultural differences exist in child outcomes following physical punishment. Some studies reported that African-American parents living in low SES areas use physical punishment as a strategy to keep their children safe, whereas European parents use physical punishment in anger. The child outcomes differ greatly between these cultures. The same may be the case for Chinese families.

59: positive parenting protects against child behaviour problems. Indeed, studies have reported this, but the ones cited here (Shek et al, 2003 and Chronis et al 2007) are not the right ones to cite. It would be better to look for studies that have a longitudinal design, specifically looking at the effect of positive parenting and not in an ADHD sample (Chronis), as the authors’ study is not about ADHD.

68: “typical problem and treatment oriented parenting programmes have limited impact at the population level.” I don’t think this is true and the cited paper doesn’t claim this.

75: I don’t understand the proposed need for shorter positive parenting programmes. Shorter than Triple P? Triple P has many different intervention levels, including very brief ones. Incredible Years is rather long, but the authors don’t refer to this programme

80: I assume to Joyful Parenting Pilot Project aimed to promote positive parenting behaviours amongst parents (not just any adult). Did this project involve several interventions?

I would like to learn a bit more about the actual intervention. How do fun family games lead to increased praise?

87: This sentence implies that only qualitative data were collected.

Methods

How were demographics obtained? Why were participants randomised by cluster? What were the inclusion and exclusion criteria? How were parents within the centres recruited?

99: “Family members (could be children or adolescents) of the participating parent were also invited to join a family gathering activity after completion of the interventions” What does this mean? Is this essential to understanding the procedure (i.e. replicating the study)?

102: “The study protocol (dated 1 June 2017) was fixed before the ethical approval and enrollment of participants.” What does this mean? Again, is this essential for the study to be replicated?

Ethical approval:

Was assent obtained from underaged children over 7 years? Clinical trial registry data is mixed up with ethical approval data. Please separate these.

Procedure:

The authors state that randomization happened via numbers generated by a computer, but continue by stating how group allocation occurred using envelopes.

The intervention procedure doesn’t make sense to me. Baseline levels were taken, measures were repeated after the first talk and again after the second talk, after which family activities took place. A more common method would be to measure baseline levels, provide the intervention to the intervention group, repeat the measures post-intervention and again at follow-up. After all assessments are done, the control group receives the intervention.

Intervention:

Describe what “Sharing, Mind and Enjoyment” and “knowledge of Mixed Anxiety and Depressive Disorder (MADD)” mean instead of just using names. Also, I would suggest rewriting this section to separate the measures (such as MADD) from the actual intervention. Be really precise in describing the intervention as this is a new intervention and readers will not have heard of it. What does interactive talk entail? It sounds like a lecture provided to parents.

Did control group parent take part in the family gathering activity? Was this event used as incentive?

Outcome measures:

It is generally better to include measures that have been validated and that are reliable, especially for primary outcome measures. With self-developed measures, you don’t know if you are measuring what you want to measure. I have strong doubts about the validity and reliability of the primary outcome measures. For the next study, I would suggesting looking at existing literature to review what kind of measures can be used to measure your outcomes of interest. This also enables comparison between studies.

In the abstract it is unclear what the purpose was of the focus group discussions and the interviews. The methods explains the purpose of the group discussions, but not of the interviews.

Also (re information in the abstract), it is more informative to know how many parents took part in a focus group discussion. It is less informative to report on the number of group discussions, as two parents can be considered a group, as well as 10. A group of 10 people will likely provide more information than a group pf two people.

Analysis of the qualitative data should not be included in the statistical analysis section.

Results

I appreciate the distinction the authors made between primary and secondary outcomes. Table one needs revision to include outcomes of the t-tests/ chi-square tests (please check APA guidelines for required content). How was the effect size calculated? Why is the effect size not included in Table 2? SD is also not reported.

I am not convinced of the results, mainly due to the self-developed primary outcome measures and the timing of the second set of assessments (during the intervention). Presentation of the results is sometimes unclear, e.g.,

263: “parents gave praise in 7.13 (SD 5.21), 6.69 (5.79), 6.63 (5.83) and 4.48 (5.84) days from the first to the fourth week” According to the methods section, the maximum score is 84. The scores presented in the results appear rather low. Also, the results should present the intervention effects, not mean scores.

I feel as if the thematic analysis could have yielded more in-depth results. If more in-depth data is available it may be interesting to present the group discussions and individual interviews in a separate paper.

Discussion

With the summary of the essential findings, it is unclear if the findings refer to the 1-month or the 3-month assessments. The authors state they expected small effect sizes, however, this was not stated in the previous sections. Why did the authors expect small effect sizes? The qualitative findings are not discussed (or I may have missed this part).

The potential theoretical and practical implications have not been identified. Apart from the suggestion to have a larger sample size, the authors do not provide suggestions for future research.

Additional comments:

As an English as second language speaker myself, I appreciate how difficult and illogical the English language can be. I would suggest to seek editorial assistance from a native English speaker. There are quite a few improvements to be made throughout the paper, e.g.

28: Two talks were delivered, including praise and appreciation skills, and enjoyable family games � this implies that “praise”, “appreciation skills”, and “enjoyable family games” are talks, while in effect these are skills taught or discussed during the talks.

30: at baseline, 1 month, and 3 months. Note: singular (and hyphen) is used in other cases, e.g. 3-month follow-up, at 3 months

31: when expressing comparisons (e.g. greater), you need to include what it is compared to, e.g. we expected greater improvement reported by intervention group parents as compared to those in the control group

39: small effect sizes (plural)

83: Praise is always singular, as is content (127)

84: easy-to-catch family games? What are those? Simple games that are easy to learn?

95 “under the Hong Kong Family Welfare Society”

99: “Family members (including children)”

Some of the references are a bit old. Some more Chinese/Hong Kong studies (Triple P):

Chan S, Leung C, Sanders M (2016) A randomized controlled trial comparing the effects of directive and non-directive parenting programs as a universal prevention program. J Child Serv 11: 38-53

Guo M, Morawska A, Sanders MR (2016) A randomized controlled trial of Group Triple P with Chinese parents in mainland China. Behav Modif 40: 825-851.

Reviewer #3: Abstract: The abstract would benefit from being more concise, particularly under methods. The hypotheses shouldn’t be stated under the methods. The authors should include information on the sample (e.g., parents of children of what age? Were this parents with concerns for their children or general public?) When did the study take place and duration? Were the focus groups conducted after or before the interventions and what were the focus groups for? The concluding statement seems to be exaggerated given the small effect size and short follow up duration. What is the impact for such a trial?

Introduction: The introduction presents Chinese parenting style. It is not clear to the reader what age group of children the authors are referring to. The introduction currently is weak and does not present a good rationale for why brief parenting programmes are needed in Hong Kong. More context should be given. The objective is to promote positive parenting practices and mental wellbeing in parents, yet no discussion is given on why parent mental wellbeing is important and how this link to children wellbeing. Line 52: Is this a belief or is there evidence to support such claim? Line 54 the author cited that more than half of Hong Kong parents has used corporal punishment. This statement requires further elaboration otherwise is out of context. The authors should state whether this data was from a survey and the sample size of the survey. Line 60: what does the author mean by children’s social achievement? Line 61-63: Not sure what the author mean in this sentence. What does the author mean by reward? Reward does not always necessarily lead to positive behaviour but can increase undesired behaviours in children. Line 68-75: Does the author mean targeted/selected parenting program? The authors should explain the public health model and why universal approaches are needed for prevention. Existing literatures should be drawn and expand on. The authors should also expand on the few studies that were conducted on Chinese parents. A stronger rationale for brief parenting programmes is needed. Line 80: What do the authors mean by pilot engagement trial?

Method: Family service centres were engaged to recruit. Were these parents already clients from the service centre? The authors stated that the family centres provide comprehensive and extensive professional support, does this mean that their service clients were those at risk or a high risk? A definition should be given re parents. Did the authors only recruited parents? What about caregivers? The inclusion and exclusion criteria should be clearly stated. Given the brief nature of the intervention, did the research team exclude families that were at risk? If so, what were used to screen families? Line 108: What about child assent? If the children did not want to participate, does that mean the family is excluded? Consort diagram should be included. More information on recruitment should also be given. Line 119: why were the assessment completed before the intervention were completed? If the 3 month assessment was conducted before the family gathering activities, how did the authors measure the effect of the family activities? The authors can consider using a diagram to show the intervention and schedule of assessments. A table can also be used to explain the content of the intervention. What was the reason for the one month gap between the two talks? Why lego? Line 149: What was the validity of the scale for measuring praise. How confident are the authors in regard to recall bias? The reliability of the enjoyment scale appear just acceptable.

Results: Line 299 There were no mention that children were required to complete assessments, this should be stated clearly in the methods section. Was specific questionnaires used to collect response from children? How reliable are data collected from a 4 year old?

Discussion: The authors state that the primary outcome is positive parenting practices, however the self-developed scales does not seem to measure parenting practices. For example, a measure on enjoyment. Can the author justify why these scales were chosen or used? It was unclear in the methods section that there were worksheet available for parents. This should be stated in the methods. The authors didn’t find significant changes in praise, and suggested that parents beliefs in the usefulness of criticism may be difficult to change in short-term. What are the implications of such findings? Perhaps modification to the content of how praise are taught? And how can the authors reduce the use of criticism in parents? Line 384: How does appreciation and enjoyment link to quality of communication? There are very few male participants, any thoughts on how this can be improved?

There are grammatical errors and typos throughout the manuscript, the manuscript would benefit from careful proof reading.

The author stated that data can not be shared due to consent reasons. Data that are deidentified should be made available upon request for example for inclusion into meta-analyses. Given journal policy for data sharing and availability, the authors should consider the inclusion of such statement in consent forms and PIS in future studies.

6. PLOS authors have the option to publish the peer review history of their article (what does this mean?). If published, this will include your full peer review and any attached files.

Reviewer #1: No

Reviewer #2: No

Reviewer #3: No

---

## [Author Response · Author response to Decision Letter 0]

7 Sep 2021

The response letter has been included as an attachment.

---

## [Decision Letter · Decision Letter 1]

24 Feb 2022

PONE-D-20-37785R1Promoting positive parenting and mental wellbeing in Hong Kong Chinese parents: a pilot cluster randomised controlled trialPLOS ONE

Dear Dr. Wang,

Thank you for submitting your manuscript to PLOS ONE. After careful consideration, we feel that it has merit but does not fully meet PLOS ONE’s publication criteria as it currently stands. Therefore, we invite you to submit a revised version of the manuscript that addresses the points raised during the review process.

The reviewers still have a number of concerns about the methodological approach and presentation of the manuscript. They also feel that improvements in the English grammar and language usage must be made. Their comments can be viewed in full, and in the attached files. Please note that for your manuscript to be considered further, each of their comments must be satisfactorily addressed.

We look forward to receiving your revised manuscript.

Kind regards,

Natasha McDonald, PhD

Associate Editor

PLOS ONE

Reviewers' comments:

Reviewer's Responses to Questions

**Comments to the Author**

1. If the authors have adequately addressed your comments raised in a previous round of review and you feel that this manuscript is now acceptable for publication, you may indicate that here to bypass the “Comments to the Author” section, enter your conflict of interest statement in the “Confidential to Editor” section, and submit your "Accept" recommendation.

Reviewer #1: (No Response)

Reviewer #2: (No Response)

2. Is the manuscript technically sound, and do the data support the conclusions?

Reviewer #1: (No Response)

Reviewer #2: Partly

3. Has the statistical analysis been performed appropriately and rigorously? 

Reviewer #1: (No Response)

Reviewer #2: Yes

4. Have the authors made all data underlying the findings in their manuscript fully available?

Reviewer #1: (No Response)

Reviewer #2: No

5. Is the manuscript presented in an intelligible fashion and written in standard English?

Reviewer #1: (No Response)

Reviewer #2: No

6. Review Comments to the Author

Reviewer #1: COMMENTS: I noted that, ABSTRACT is well drafted but assay type. Please note that it is preferable to divide the ABSTRACT with small sections like ‘Objective(s)’, ‘Methods’, ‘Results’, ‘Conclusions’, etc. which is an accepted practice of most of the good/standard journals [including this one]. If I remember correctly, it was divided in small sections earlier. May please consider (again).

Thank you very much for adding a word “pilot” in title and removing ‘p-value’ column in Table 1 [i.e. statistical comparison of baseline Characteristics] as suggested {both actions highly appreciated}. I completely agree that “the cluster effect should be adjusted for and thus the Mann-Whitney ‘U’ test on ‘change scores’ may not be usable”, but wanted to indicate that direct / head-to-head comparison between groups is desirable [that does not mean that use of “Multilevel mixed-effects linear regression model” which definitely adjust for clustering effect is wrong but note that this technique is not originally developed for comparison of between-group mean differences. In this context [authors may already know the reference], here is a good reference: Donner Allen and Klar Neil. `Design and Analysis of Cluster Randomization Trial in Health Research’, Oxford University Press Inc., New York, 2000.

The fact that ‘though the measures/tools used here are appropriate, most of them yield data that are in ‘ordinal’ level of measurement [and not in ratio level of measurement for sure {as the score two times higher does not indicate presence of that parameter/phenomenon as double (for example, a Visual Analogue Scales VAS score or say ‘depression’ score)}].

I certainly know the vital importance of ‘Focus groups and in-depth interviews’ in research but I only suggested to reduce account given in section.

Ultimately, in my considered opinion now, ‘let the respected editor decide the future course’. I do not have any specific recommendation.

Reviewer #2: The authors put in a lot of effort to take on board reviewers’ suggestions. However, there are still outstanding issues that need to be addressed, plus some new ones due to the changes in the introduction. There are still quite a few English grammar mistakes. E.g.,

line 74: “meaning to govern and train but also to show care and love”, line 78 “reported themselves as low acceptance”,

line 83 “a latest report”. Please make use of a proficient English speaker for the next version.

Line 111: “Chinese parents showing warmth and reward to children is more of getting children’s”

Line 173: “Those who cannot read Chinese or were suffering from “

My comments relate to the revision with the track and changes.

Abstract:

The abstract could benefit from some reorganising. It is helpful to stick to the common “intro, methods, results, discussion, conclusion”. Be succinct and only state the essential.

Line 35: Please add children’s mean age.

Line 38: I am unclear what the difference may be between praise and appreciation skills.

Line 39: remove the comment on the control group not receiving an intervention during the study. This is how control groups work. Instead, you could state, “the control group participants were offered the intervention after all data were collected”

I would advise to make a distinction between primary outcomes and secondary outcomes in the results reported in the abstract as well.

Introduction:

The authors made the introduction more relevant. There is still some restructuring and fine tuning needed. E.g.,

Line 79: The authors state “despite cultural differences in training and controlling”. However, these differences have not been discussed, it was only stated that training and control is considered positive in China.

Line 77: Sentence structure is incorrect. Western parents use praise to boost children’s confidence, whereas Chinese parents use praise to ..” or “Western parents tend to use praise in abundance in order to boost their children’s confidence, whereas Chinese parents tend to use praise sparingly to prevent their children becoming arrogant (NB arrogant is just an example).

Line 82: Unclear what is meant, please clarify. Is this what the authors meant: “.. more than half of HK parents had used corporal punishment, with 4.5% of these cases resulting in injuries”

The authors sometimes jump from one topic to the next and the research that is cited does not always apply to the arguments being made. E.g., after talking about physical abuse (until line 89), the authors suddenly start discussing social support.

Line 92: why do the authors talk about students? What is students’ life satisfaction? Children’s satisfaction with their life at school? This can have so many other causes other than parenting. How old were these children?

Again, the use of ‘students being satisfied’ in line 107. Maybe students were not satisfied because of the amount of homework. This needs to be better specified. The following suggestion regarding family support (line 108) implies that the students in line 107 had received family support.

Line 110: “yet parents may not know how to praise children appropriately”. Earlier in the introduction the authors talked about praise and parents preferring to be strict. This needs to be combined and some additional literature needs to be reviewed. Perhaps this can be combined with the parenting challenges that Chinese parents experience.

Lines 113-117: Good addition of praise types. Please add an example of process praise.

Line 127-128: references are needed to back up these statements

Lines 129-133: the authors discuss Triple P as a good example of a universal parenting intervention. This makes the reader wonder why the authors did not use that programme instead, which is well known, effective, and translated into Mandarin. What is the rationale for using the authors’ intervention? With 4 group sessions, Triple P is already pretty brief. Is it too costly (i.e., you do need a qualified triple P facilitator)? Are there too many points of contact with triple P (i.e., 4 group sessions vs only 2 interactive talks)?

For attachment-based interventions research suggests that shorter is better. No such evidence exists for behavioural-based interventions. So why is shorter better? It seems to me that the authors’ intervention is time consuming too, with all the family games to be played over the course of one (or two?) months.

Line 145: to what extent were parents involved in the study design? I see no evidence of CBPR in the current study except for service providers being interviewed. This is not participatory research.

Methods

Thanks for providing a rationale for using a clustered design.

Line 196: leave out: “The intervention group received the interventions first.” It is sufficient to say that the control group received access to the intervention after data collection was complete.

Line 197: the term follow-up is not appropriate here as there is no actual follow-up assessment (i.e., assessment after the post-intervention assessment). Common description of assessment points are: baseline (Time 1), post-intervention (Time 2), ..-month follow-up (Time 3). Not sure how to describe the authors’ research design, as Time 2 is only after the first part of the intervention.

Line 203: leave out: “.. no interventions during the assessment period” see my previous point re line 196.

Joyful parenting intervention/ SME intervention lines 205-248 - I find the use of both these names confusing

- What is the name of the actual intervention?

- Who developed this and how?

- Based on what?

- What is the interactive aspect of the talks? They still sound like lectures to me.

- Is there information available to the public so this study can be replicated?

- Line 210: praise (positive sharing). The definition of ‘praise’ is not ‘positive sharing’. I think most readers will know what praise is (the expression of approval), so perhaps not necessary to supply a definition here.

- Regarding ‘Appreciation skills’, I have only ever heard of literary appreciation skills, so not sure what this refers to.

- Line 236: family games during 4 or 3 weeks? The authors talk about 4 weeks, but then state 21 days.

- How much paly and practice happened? One or two months?

Line 217, what does MADD stand for (this has been deleted) – acronyms should be explained at first mentioning.

Lines 219-220 (reference 29) belongs in the introduction.

Lines 226-232: this belongs in the introduction, not in the methods. Please integrate with what is already in the introduction.

The goal of the methods section is to allow for a study to be replicated by someone else. This is currently not possible and more information needs to be provided.

No evidence of any interaction in the ‘interactive talks’. These talks sound like lectures and should not be described as being interactive. The subsequent games do seem interactive as do the practice with focussing on positive traits

There is insufficient evidence of the validity and reliability of the primary outcome measures. How do the authors know their measures assess what they were designed to assess? Any changes found between the control and intervention group could be something else entirely. Retrospective behaviour-related questions (e.g., to quantify the amount of praise) are not necessarily reliable. Plus, as there was no placebo intervention, any changes measured may be due to the effect of attention. These are all aspects that need to be discussed in the limitation section of the study.

The high level of correlation between the different primary outcomes is not necessarily positive as it may be indicative of the different measures assessing the same construct rather than separate constructs. Evidence on some of the secondary outcomes is much stronger as some were based on existing questionnaires.

Measuring children’s perception of the level of parental praise is a great way to make the parent measure of praise stronger. This should be emphasised more, but only if the child questions were very similar to the parent questions.

Fidelity only checked by the researchers, not by an independent other. How was the quality of the qualitative component ensured?

Discussion

Line 503: “.. two simple interactive talks”. What about all the family games that took place over the course of one (or two?) months? It’s not such a brief intervention after all. There are only two points of contact.

7. PLOS authors have the option to publish the peer review history of their article (what does this mean?). If published, this will include your full peer review and any attached files.

Reviewer #1: No

Reviewer #2: No

---

## [Author Response · Author response to Decision Letter 1]

2 Apr 2022

The response letter to the reviewers has been attached.

---

## [Decision Letter · Decision Letter 2]

28 Apr 2022

PONE-D-20-37785R2Promoting positive parenting and mental wellbeing in Hong Kong Chinese parents: a pilot cluster randomised controlled trialPLOS ONE

Dear Dr. Wang,

Thank you for submitting your manuscript to PLOS ONE. After careful consideration, we feel that it has merit but does not fully meet PLOS ONE’s publication criteria as it currently stands. Therefore, we invite you to submit a revised version of the manuscript that addresses the points raised during the review process.

The manuscript has been evaluated by two reviewers, and their comments are available below.

The reviewers have raised some of concerns that need attention. They request additional information on methodological and reporting aspects of your study.

Could you please revise the manuscript to carefully address the concerns raised?

We look forward to receiving your revised manuscript.

Kind regards,

Thomas Phillips, PhD

Staff Editor

PLOS ONE

Journal Requirements:

Reviewers' comments:

Reviewer's Responses to Questions

**Comments to the Author**

1. If the authors have adequately addressed your comments raised in a previous round of review and you feel that this manuscript is now acceptable for publication, you may indicate that here to bypass the “Comments to the Author” section, enter your conflict of interest statement in the “Confidential to Editor” section, and submit your "Accept" recommendation.

Reviewer #1: All comments have been addressed

Reviewer #2: All comments have been addressed

2. Is the manuscript technically sound, and do the data support the conclusions?

Reviewer #1: (No Response)

Reviewer #2: Yes

3. Has the statistical analysis been performed appropriately and rigorously? 

Reviewer #1: (No Response)

Reviewer #2: Yes

4. Have the authors made all data underlying the findings in their manuscript fully available?

Reviewer #1: (No Response)

Reviewer #2: No

5. Is the manuscript presented in an intelligible fashion and written in standard English?

Reviewer #1: (No Response)

Reviewer #2: Yes

6. Review Comments to the Author

Reviewer #1: COMMENTS: Since all of the comments made on earlier draft by me (and hopefully by other respected reviewers also) were/are attended positively, I recommend the acceptance because the manuscript now has achieved acceptable level, in my opinion.

Reviewer #2: The authors did a fantastic job with the revisions and with taking on board all reviewers’ suggestions. The paper has improved immensely over the course of the revision rounds.

I just have a few comments (relating to the revised manuscript with the visible track changes). I do not need to review any revisions.

In the CONSORT diagram it says that all centres and parents allocated to the control group received the intervention. I find this confusing as the control group was a care-as-usual group and only received the intervention after data collection was complete.

line 93-101:” Parents may benefit from training” would be a more nuanced statement instead of “Parents need training”. Also, if talking about substantial proportions, some kind of evidence on these proportions must be provided. The authors listed some of the stressors parents may have, but not how many parents experience this. Perhaps the easiest way to deal with this is to rephrase the leading sentence in line 93 and not use the words “substantial proportion”.

lines 123-128 – excellent addition to explain the campaign. Perhaps put the words sharing, mind, and enjoyment in between apostrophes, e.g., ‘Sharing’ refers to connecting with family and friends. Because sharing means something else in ‘everyday English’.

7. PLOS authors have the option to publish the peer review history of their article (what does this mean?). If published, this will include your full peer review and any attached files.

Reviewer #1: **Yes: **Dr. Sanjeev Sarmukaddam

Reviewer #2: **Yes: **Nike Franke

---

## [Author Response · Author response to Decision Letter 2]

2 May 2022

The response letter has been attached.

---

## [Decision Letter · Decision Letter 3]

3 Jun 2022

Promoting positive parenting and mental wellbeing in Hong Kong Chinese parents: a pilot cluster randomised controlled trial

PONE-D-20-37785R3

Dear Dr. Man Ping Wang,

We’re pleased to inform you that your manuscript has been judged scientifically suitable for publication and will be formally accepted for publication once it meets all outstanding technical requirements.

Kind regards,

Yann Benetreau, PhD

Division Editor (Staff Editor)

PLOS ONE

Additional Editor Comments (optional):

Reviewers' comments:

Reviewer's Responses to Questions

**Comments to the Author**

1. If the authors have adequately addressed your comments raised in a previous round of review and you feel that this manuscript is now acceptable for publication, you may indicate that here to bypass the “Comments to the Author” section, enter your conflict of interest statement in the “Confidential to Editor” section, and submit your "Accept" recommendation.

Reviewer #1: All comments have been addressed

Reviewer #2: All comments have been addressed

2. Is the manuscript technically sound, and do the data support the conclusions?

Reviewer #1: (No Response)

Reviewer #2: Yes

3. Has the statistical analysis been performed appropriately and rigorously? 

Reviewer #1: (No Response)

Reviewer #2: Yes

4. Have the authors made all data underlying the findings in their manuscript fully available?

Reviewer #1: (No Response)

Reviewer #2: No

5. Is the manuscript presented in an intelligible fashion and written in standard English?

Reviewer #1: (No Response)

Reviewer #2: Yes

6. Review Comments to the Author

Reviewer #1: COMMENTS: Since all of the comments made on earlier draft by me (and hopefully by other respected reviewers also) were/are attended positively, I recommend the acceptance because the manuscript now has achieved acceptable level, in my opinion.

Reviewer #2: (No Response)

7. PLOS authors have the option to publish the peer review history of their article (what does this mean?). If published, this will include your full peer review and any attached files.

Reviewer #1: **Yes: **Dr. Sanjeev Sarmukaddam

Reviewer #2: **Yes: **Nike Franke

---

## [Editor Report · Acceptance letter]

11 Jul 2022

PONE-D-20-37785R3 

Promoting positive parenting and mental wellbeing in Hong Kong Chinese parents: a pilot cluster randomised controlled trial 

Dear Dr. Wang:

I'm pleased to inform you that your manuscript has been deemed suitable for publication in PLOS ONE. Congratulations! Your manuscript is now with our production department. 

Kind regards, 

on behalf of

Dr. Yann Benetreau 

Staff Editor

PLOS ONE